# Synthesis, Characterization and Fabrication of Graphene/Boron Nitride Nanosheets Heterostructure Tunneling Devices

**DOI:** 10.3390/nano9070925

**Published:** 2019-06-27

**Authors:** Muhammad Sajjad, Vladimir Makarov, Frank Mendoza, Muhammad S. Sultan, Ali Aldalbahi, Peter X. Feng, Wojciech M. Jadwisienczak, Brad R. Weiner, Gerardo Morell

**Affiliations:** 1Department of Physics, Engineering and Astronomy, Austin Peay State University, Clarksville, TN 37040, USA; 2Institute for Functional Nanomaterials, University of Puerto Rico, San Juan Puerto Rico, PR 00936, USA; 3Department of Physics, University of Puerto Rico, San Juan Puerto Rico, PR 00936, USA; 4King Abdullah Istitute for Nanotechnology, Department of Chemistry, College of Science, King Saud University, Riyadh 11451, Saudi Arabia; 5School of Electrical Engineering and Computer Science, Ohio University, Athens, Ohio, OH 45701-2979, USA; 6Department of Chemistry, University of Puerto Rico, San Juan Puerto Rico, PR 00936, USA

**Keywords:** graphene, boron nitride nanosheets, heterostructures, tunneling device, 2D materials

## Abstract

Various types of 2D/2D prototype devices based on graphene (G) and boron nitride nanosheets (BNNS) were fabricated to study the charge tunneling phenomenon pertinent to vertical transistors for digital and high frequency electronics. Specifically, G/BNNS/metal, G/SiO_2_, and G/BNNS/SiO_2_ heterostructures were investigated under direct current (DC-bias) conditions at room temperature. Bilayer graphene and BNNS were grown separately and transferred subsequently onto the substrates to fabricate 2D device architectures. High-resolution transmission electron microscopy confirmed the bilayer graphene structure and few layer BNNS sheets having a hexagonal B_3_-N_3_ lattice. The current vs voltage I(V) data for the G/BNNS/Metal devices show Schottky barrier characteristics with very low forward voltage drop, Fowler-Nordheim behavior, and 10^−4^ Ω/sq. sheet resistance. This result is ascribed to the combination of fast electron transport within graphene grains and out-of-plane tunneling in BNNS that circumvents grain boundary resistance. A theoretical model based on electron tunneling is used to qualitatively describe the behavior of the 2D G/BNNS/metal devices.

## 1. Introduction

Graphene [1] and other atomically thin layered materials [2,3] have opened up new possibilities for fabricating compact 2-dimensional (2D) nanomaterial heterostructures with novel electronic and optoelectronic properties. Such architectures have the potential to enable new electronic devices and energy storage media. Stacking graphene on boron nitride nanosheets (BNNS) is one example [4] of such 2D/2D device structures useful particularly for vertical transistors in digital and high frequency electronics [5]. There is a promise for 2D layered materials to yield new device heterostructures that can work in low power and high frequency electronics without significant change in input characteristics.

Although free-standing graphene has shown unique and outstanding mechanical and electronic properties, however when it is in contact with a substrate surface its physical properties change due in part to parasitic transport effects. Furthermore, the high carrier mobility of graphene is drastically reduced by grain boundary electron scattering and substrate interactions that disturb the charge distribution, thus limiting its immediate device applications.

BNNS is an emerging 2D material with similar crystal structure to that of graphene (G) and a small lattice mismatch of about 1.45% that makes it a strong candidate for developing graphene-based heterostructures [6]. It is a wide bandgap material having a hexagonal network of alternating B and N atoms. It offers relatively large and smooth surfaces free of dangling bonds, hence, electronic perturbations when in contact with graphene should be minimal. A sheet of BNNS comprised of few atomic layers offers a unique set of physical properties [7,8,9,10,11], including atomically thick electron tunneling barrier, which is of particular interest for devices based on layered materials [12].

Nanothin BN, such as our BNNS, shows in-plane insulating properties and out-of-plane charge tunneling mechanism that contributes in the conductivity of the material underneath [13]. The out-of-plane electron tunneling through atomically thin layers of BNNS makes this material advantageous for 2D/2D heterostructure devices where BNNS can be used as gate-controlled p-layer [13]. There are reports [14,15] indicating that the physical properties of graphene improve significantly when it is brought into physical contact with BNNS to develop a heterostructure. Therefore, the quality and performance of G/BNNS interface as well as the effect of substrates (i.e., conducting vs. non-conducting) on heterostructure performance is of particular interest. It is expected that the polarity of the B–N bond results in interlayer electrostatic interactions with the C atoms residing on the underlying B or N atoms, thus stabilizing the AA’ stacking mode [4]. Moreover, these types of electrostatic interactions play a role in the interlayer bonding between different layered materials. Specifically, the electrostatic attractions between adjacent BNNS layers and graphene sheets are expected to reduce the interlayer distance similarly to the case of graphene deposited on silicon oxide substrates. It is worth noting that the combination of G/BNNS heterostructures on different conducting and non-conducting substrates also affects the properties of the system [16]. While studying the G/BNNS/metal (G/BNNS/M) heterostructures, the significance of the out-of-plane charge tunneling mechanism is carefully considered in this report. The results are explained using a charge tunneling model, where the barrier is dependent on the applied electric potential difference.

## 2. Experimental Materials and Procedures

There are detailed reports on the synthesis of bilayer graphene [17] and BNNS sheets using chemical vapor deposition (CVD) techniques [18,19]. In this study, two different synthesis techniques were employed to synthesize reliable, reproducible 2D layered materials: CO_2_-pulsed laser deposition (CO_2_-PLD) and hot filament chemical vapor deposition (HFCVD; Hot Filament CVD Instrument, Blue Wave Semiconductor Inc, Baltimore, MD 21227, USA) for BNNS and graphene, respectively. The BNNS synthesis process was carried out by irradiating a pyrolytic *h*-BN target using CO_2_-PLD (GSI Lumonics Inc., 105 Schneider Road, Kanata (Ottawa) Ontario, Canada) operating at 10.6 µm with a pulse width of 1-5 µs, repetition rate of 5 Hz, and pulse energy of 5 J. Copper foil (2 × 2 sq. inch), molybdenum disc (1″ diam.), and highly polished SiO_2_ (1″ diam.) were used as substrates. Substrates were mounted on a holder 3 cm away from the target. A detailed description of the CO_2_-PLD synthesis has been reported previously [19,20,21,22]. HFCVD was applied for graphene synthesis [17]. To fabricate the 2D device architectures, the synthesized graphene samples were transferred mechanically on the surface of BNNS or SiO_2_ using the polymethyl methacrylate (PMMA) technique, frequently used to transfer graphene between different substrates [17]. First, a layer of PMMA solution was spin-coated onto the graphene layer to act as a support. Next an etchant was used to remove the copper substrate leaving the PMMA/graphene layers stack ready to be transferred to BNNS and SiO_2,_ respectively. Finally, the PMMA layer was etched away by dissolving it with acetone leaving behind a G/BNNS heterostructures. The surface morphology and crystal structure of the graphene sheets were analyzed using scanning electron microscope (SEM; JEOL JSM-7500F, Akishima, Tokyo 196-8558, Japan), high-resolution transmission electron microscope (HRTEM; Spherical aberration corrected high resolution transmission electron microscopy, Titan), and Raman spectroscopy (Thermo Scientific DXR Confocal Raman Microscope; Thermo Electron North America LLC, West Palm Beach, FL 33407, USA) equipped with optical microscopy accessories. The quality of the graphene sheets was verified at various stages along the process using Raman spectroscopy: As-grown, on PMMA, and after transfer to SiO_2_ and BNNS. The devices that ended up with poor quality graphene sheets due to damage/contamination during the transfer process were discarded. For device prototyping and characterization, the sputtering technique was used to deposit metal contacts (Au) on the graphene surface side of the devices. The contacts were 100-nm thick and approximately 1 mm apart from each other. The thickness of BNNS nanolayers was estimated using the XRD method and using an algorithm reported by M. Yasaka [23].

The I(V) data of the 2D/2D devices were obtained by the Van der Paw [24] method in which the roles of the electrodes were systematically interchanged in order to verify reproducibility and calculate average values with standard deviation (see Table 1). In our study we tested four devices independently. Three devices were composed of G/BNNS/M with different thickness of BNNS, 10 nm, 100 nm, and a few microns of BNNS sandwiched between graphene and metal substrate. Two devices with BNNS 10 nm and 100 nm have shown excellent I(V) behavior and data is presented in current paper. The device prepared with few micron BNNS thickness (~6 µm) did not show IV characteristics due to large barrier thickness, and lack of tunneling mechanism. Similar situation was observed in the case of G/BNNS/SiO_2_ and we also did not present data in our study. Reason is G/BNNS/SiO_2_ did not show I(V) characteristics due to the high resistance of the device as two non-conducting surfaces were in contact underneath of the graphene layer. Moreover, at least three devices of each type were fabricated and tested to ensure reproducibility. All experiments were conducted at room temperature.

## 3. Results and Discussion

The surface morphology and nanoscale structures of as-synthesized BNNS and graphene sheets were carefully evaluated. The BNNS images collected by scanning electron microscope (SEM) are presented in Figure 1a,b. The micrographs show smooth and flat BNNS (Figure 1a). Figure 1b shows the BNNS at larger length scale where individual layers were seen beneath the top nanosheet. Figure 1c shows the corresponding TEM image of the transparent BNNS large area. To analyze the BNNS morphology at the nanoscale, we focused the electron beam at the edge and on the surface of the selected BNNS. The HRTEM image of the edge is shown in Figure 1d depicting the highly crystalline layer-by-layer structure having excellent interfaces. The high-resolution image (Figure 1e) shows the hexagonal lattice structure.

To obtain high-resolution images we operated the TEM at 200 kV. Using such high acceleration potential produced the irradiation damages and defects in the BNNS due to a knock-on effect seen in Figure 1e. The knock-on damage threshold for B and N atoms in BNNS is about 74 kV (B) and 84 kV (N), respectively, and is slightly lower than that of C atoms in graphene [19]. Some areas in the image (Figure 1e) still show the hexagonal lattice of B_3_–N_3_ atoms within the sheet as shown in the inset with lattice constant 0.22 Å. Figure 1f shows the magnified image of a perfectly stacked layer-by-layer BNNS structure at an atomic scale with spacing 3.34 Å. We believe that sheets having such perfect stacking of layered structures could provide a suitable platform for graphene to produce a good heterostructure. The selected area electron diffraction pattern in Figure 1g shows bright dots indicating the polycrystalline nature of BNNS sheets and their hexagonal B_3_–N_3_ structure.

The identification of graphene layers under the optical microscope is a challenging task due to its highly transparent nature, thickness down to two atomic layers and weak contrast with the substrate. In order to overcome this challenge, we mechanically transferred the graphene sheets onto quartz substrates and collected optical images when illuminated at 600 nm. Figure 2a shows schematically the transfer process of a graphene sample using the PMMA technique onto selected substrates. Since 2D graphene layers are highly flexible we observed the layers folding. Despite that, we were able to identify the graphene sheet large area sample with existing folding seen at a few locations as shown in Figure 2b. We identified a few different location markers as 1 to 3 in Figure 2a and collected the corresponding Raman spectra as shown in Figure 2c. The Raman spectra shows bands characteristic of bilayer graphene with an estimated I_2D_/I_G_ ratio >1 [17]. The Raman intensity increased from location 1 to location 3. It is seen in Figure 2b that the graphene layer at location 1 was fold free, whereas number of folds increased at location 2 (one fold) and at location 3 (two folds), respectively. It is interesting to note that the folded locations were still showing the Raman signatures for bilayer graphene whereas the peaks’ intensity gradually increased with the number of folds. We believed that folded graphene layers induced stress, which in turn caused the honeycomb rings to partially elongate causing a slight change in the lattice constant towards the direction of the force. At the same time, the distance between two consecutive layers was reduced. As a result of that, the graphene layer was more sensitive to perturbation, e.g., phonon–phonon vibrations within the layer plane as well as between the layer interface. Figure 2d shows a representative HRTEM image of graphene transferred onto the TEM grid. The TEM analyses confirmed the conclusions drawn from the Raman study. It is clearly seen that graphene sample was a bilayer with individual graphene layers displaced with respect to each other as marked by arrows in Figure 2d.

We then developed prototype 2D/2D devices by fabricating G/SiO_2_, G/BNNS/SiO_2_, and G/BNNS/M heterostructures. Figure 3a shows the schematics of a typical device configuration tested. The performance of each device was assessed by measuring current vs. voltage I(V) and estimating the corresponding sheet resistance of graphene (see Table 1) using the Van der Paw method [24]. The sample holder with multiple devices was mounted on the four-probe holder for collecting the I(V) data at room temperature. It was found that for the G/SiO_2_ heterostructure the sheet resistance was about 10^3^ Ω/sq, which increased three orders of magnitude to about 10^6^ Ω/sq for G/BNNS/SiO_2_. The high sheet resistance observed for these heterostructures was due to the extensive presence of grain boundaries (50–70 nm grain sizes). Moreover, the lattice mismatch between the hetero-materials as well as overall thickness of the insulating layers contributed together to suppressing any possibility for out-of-plane tunneling effects.

In contrast to the G/SiO_2_ and G/BNNS/SiO_2_, the I(V) characteristics of G/BNNS/M heterostructures show good conductivity. The sheet resistance of this device is seven orders of magnitude lower than that of G/SiO_2_ down to 10^-4^ Ω/sq. The low sheet resistance of this particular configuration may be understood as arising from BNNS charge tunneling mechanism in the vertical direction (i.e., out-of-plane) when it is sandwiched between two conductive materials [13,25,26]. When the locally enhanced electric field overcomes the bandgap energy of BN (~6 eV), one can expect transport of electrons to the conduction band and concomitant faster electron transport between the electrodes on the graphene layer. This implies that the charge transport mechanism through BNNS perpendicular to the plane is about as high as the in-plane electron conductivity of graphene [14]. If this explanation is correct, one may assert that the G/BNNS/M heterostructure helps to circumvent grain boundary resistance in graphene layers, thus drastically reducing the effective sheet resistance. Hence, such property of 2D/2D graphene/BNNS may be useful for improving the performance of 2D architectures for flexible electronics.

The I(V) characteristic measurements conducted for G/BNNS/M having 10 nm (Curve 1) and 100 nm (Curve 2) thickness are shown in Figure 3b. Each individual device was tested for I(V) characteristic multiple times by completing the circuit by touching external metal probes at different locations on the Au metal contact pads. Measured I(V) characteristics are well reproducible with 7% standard deviation between individual I(V) measurement for a tested device.

The performance of each individual device was analyzed by measuring electrical current vs. voltage I(V) characteristics between two individual metal contacts and by estimating the corresponding sheet resistance of G layer using Van der Pauw method [24]. The carrier with multiple devices was mounted on the four-probe holder for collecting the I(V) characteristics at 300 K. The tests conducted for G/SiO_2_, G/BNNS/SiO_2_ showed no I(V) characteristics due to the high resistance of the devices. However, the measurements conducted for G/BNNS/M showed I(V) characteristics of a typical Schottky diode [20] with low forward voltage drop as represented by Curve 1 in Figure 3b. A significant shift in the I(V) curves was observed when measurements were collected from G/BNNS/M devices having thicker BN layer thickness (see Curve 2 in Figure 3b). The observed voltage shift seen in those devices, we believed, was due to the interface and barrier inhomogeneities between the G and BNNS layers. Shift of second curve with respect to first one was about 2.5 × 10^−5^ V. The Dc power source used for measurement offered uncertainty better than 0.1 × 10^−5^ V, which was sufficient to get correct measurement in both cases. It is worth noticing that in our previous study on BNNS/M device [19] we observed a forward voltage drop about 35 V. However, in the present study on G/BNNS/M devices the forward voltage drop was drastically reduced. Thus, we believed that such a significant voltage change occurred due to the high electrical conductivity of the G layer serving as an electrode in the tested G/BNNS/M device [26].

The observed decrease in current with an increase of BNNS film thickness (Figure 3b) was analyzed using the Fowler-Nordheim formalism finding that it was consistent with the electron tunneling transport mechanism described above. To further understand these experimental results, we developed a theoretical model of charge tunneling across a thin insulating layer sandwiched between two conducting layers. The model relays on local quantum tunneling effects [27,28] and qualitatively describes the collected experimental data. To explain the conductivity effect observed in the G/BNNS/M heterostructure, we first discussed the potential between BNNS surface and the metal (M) interface. These two potentials are schematically shown in Figure 4a as a rectangular box potential (M) and a rectangular potential barrier (BNNS) where U_0_ is the potential barrier height and ϕ_M_ is the work function of metal, respectively. Figure 4b shows the BNNS potential barrier width gradual change with external voltage V applied to the probing electrode metal (PEM) deposited on the graphene layer causing an increase of the overall electrons tunneling probability through the barrier. Here the *a*_BN_ parameter represents the barrier width and A is the layer critical thickness. We assume that the effect of applied *V* was to round off the corner of the potential barrier U_0_ and to narrow the average barrier width, as shown in Figure 4b, such that it would initiate the electron tunneling process.

Numerical calculations of the resulting tunneling current were carried out for the general system shown in Figure 4b, where the tunneling current is defined as [29],
(1)$$I_{tun}\left(Ε\right)\approx n_{0}\left|e\right|\int_{0}^{U_{0}}e^{-\frac{2\sqrt{2m^{*}}}{h\alpha \left(Ε\right)}{\left(U_{0}-E_{e,met}\right)}^{\frac{3}{2}}-\frac{E_{el,met}}{k_{B}T}}dE_{e,met},$$
where *n*_0_ is the number density of the conductive electrons in the metal, |*e*| is the absolute value of the electron charge, *m^*^* is the effective electron mass, *h* is the Plank constant, *α*(**E**) is the parameter dependent of an applied electric potential difference and increases with an increase of potential difference (dimension is (*eV/nm*)), U_0_ is the potential barrier amplitude, *E_e,met_* is the energy of the conductive electrons in the metal, *k_B_* is Boltzmann constant, and *T* is temperature. To carry out the numerical analysis of interest, we assumed that *α*(**E**) = *a***E**, where **E** is the electric field strength developed in the sample of interest. In our case, such field was defined as *V/s*, where *V* is the applied potential difference and *s* is the thin film thickness. The *U*_0_ value can be assigned with good accuracy to the BN bandgap (~6 eV [30]). Here, the *a* parameter is dependent only on the thin film nature. Thus, carrying out a fitting procedure using Equation (1) for data shown in Figure 3b, we estimated the values of the *a* and *U*_0_ parameters to be 2.71 ± 0.48 (*eV/nm*) × (*cm/V*) and 5.7 ± 0.6 eV, respectively. This value is the result of averaging of *a* parameter values obtained by fitting procedure for both curves shown in Figure 3b. To check the model sensitivity to *U*_0_ variation around the ionization threshold of BN sheets, the tunneling current dependence on thin film thickness was simulated, where analysis was carried out for the average interval of the applied potential difference used in the current study. Results for the tunneling current through the G/BNNS heterostructure analysis described by Equation (1) are presented in Figure 4. It is seen that with an increase of the BNNS layer thickness, the tunneling current asymptotically approached zero. This implies that the potential barrier induced by BNNS layer thickness controlled the electron transport between the two conducting channels. The tunneling current through a fixed BNNS layer thickness increased when the applied voltage increased due to the dependence on the potential barrier profile. We believed that BNNS thickness and potential barrier thickness parameters simultaneously affected the tunneling current. Results shown in Figure 5 were normalized to the maximum value of the relative tunneling current obtained for 25 nm think potential barrier and U = 10 eV, respectively. Taking into account the BNNS thickness (10–20 nm) and the selected parameter values, we estimated that the probability of electron tunneling for the average applied potential difference was in the range of 0.22–0.53 for the hereby considered BNNS thicknesses.

Notice that the above consideration neglects the thermionic current (i.e.*,* thermo emitted charges passing above barrier) that can be approximated by the relation:(2)$$I_{therm}\left(Ε,T\right)\approx n_{0}\left|e\right|\int_{U_{0}}^{\infty }e^{-\frac{E_{el,met}-\left|e\right|\delta hΕ}{k_{B}T}}dE_{e,met}=n_{0}\left|e\right|k_{B}Te^{-\frac{U_{0}-\left|e\right|V}{k_{B}T}}.$$
When considering Equation (2) with U_0_ ~6 eV (i.e., BN bandgap) and the applied potential difference in the mV range, it is justified to assume that the thermionic emission is small and can be neglected in the case studied. Therefore, we assumed that the main mechanism of electron transport through the BN layer in the G/BNNS/M heterostructures was only tunneling. We could thus state that the I(V) experimental data of G/BNNS/M were properly accounted for by the tunneling charge transport mechanism. When we substituted the metal substrate for an insulating substrate the results changed dramatically. Figure 6 shows the potential profiles schemes for G/BNNS/SiO_2_ (Figure 6a) and PEM/G/BNNS/SiO_2_ (Figure 6b), where U_1_ is the potential barrier of SiO_2_ and U_0_ is the potential barrier of BNNS, respectively. In the case of the PEM/G/BNNS/SiO_2_ system, by placing two insulating sheets, there was high sheet resistance and no tunneling current out of the plane should occur. In fact, these simulated results corresponded well to the experimentally observed values (see Table 1). Therefore, the model indicates that the conductivity of the G/BNNS/M systems had a considerable contribution from tunneling phenomena, while the PEM/G/BNNS/SiO_2_ system prevented tunneling.

## 4. Conclusions

Bilayer graphene and BNNS were grown separately and transferred to fabricate 2D device architectures: G/BNNS/Metal, G/SiO_2_, and G/BNNS/SiO_2_. These heterostructures were investigated under DC-bias conditions at room temperature. The I(V) data for the G/SiO_2_ and G/BNNS/SiO_2_ devices show very high sheet resistances, while the G/BNNS/Metal devices show 10^−4^ Ω/sq sheet resistance and Fowler-Nordheim behavior. This result is explained as the combination of fast electron transport within graphene grains and out-of-plane tunneling in BNNS that circumvents grain boundary resistance. A theoretical model based on electron tunneling was used to qualitatively describe the behavior of the 2D G/BNNS/Metal devices. Taken altogether, the results hereby presented indicate that G/BNNS/M heterostructures might help to circumvent grain boundary resistance in graphene layers by drastically reducing the effective sheet resistance. This finding may be useful for improving the performance of 2D device architectures.

## Figures and Tables

**Figure 1 nanomaterials-09-00925-f001:**
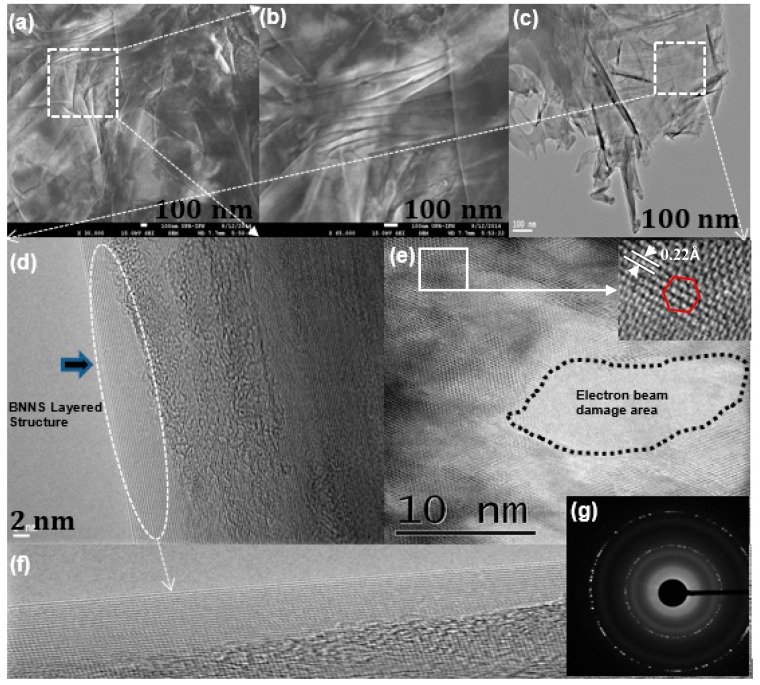
Electron microscopy characterization of boron nitride nanosheets (BNNS). (**a**,**b**) High-resolution plane view SEM images; (**c**) low magnification TEM image; (**d**–**f**) high-resolution transmission electron microscope (HRTEM) image and magnified areas of the selected BNNS. Insert in (**e**) confirms that the hexagonal BN structure is identifiable at the atomic scale with lattice constant 0.22Å. (**g**) Selected area electron diffraction pattern indicating polycrystalline nature of BNNS.

**Figure 2 nanomaterials-09-00925-f002:**
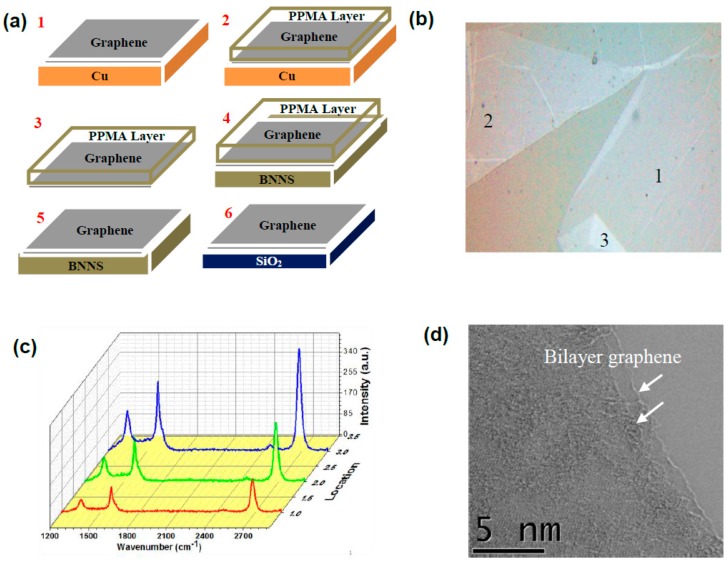
(**a**) Schematic diagram illustrating graphene transfer procedure from copper substrate to BNNS and SiO_2_ supports using PMMA process. (**b**) Optical image of the graphene sheet transferred on quartz substrate under 600 nm illumination collected by Raman system. (**c**) Raman spectra collected from different locations marked in (**b**). (**d**) HRTEM image showing graphene bilayer with observed displacement between individual layers.

**Figure 3 nanomaterials-09-00925-f003:**
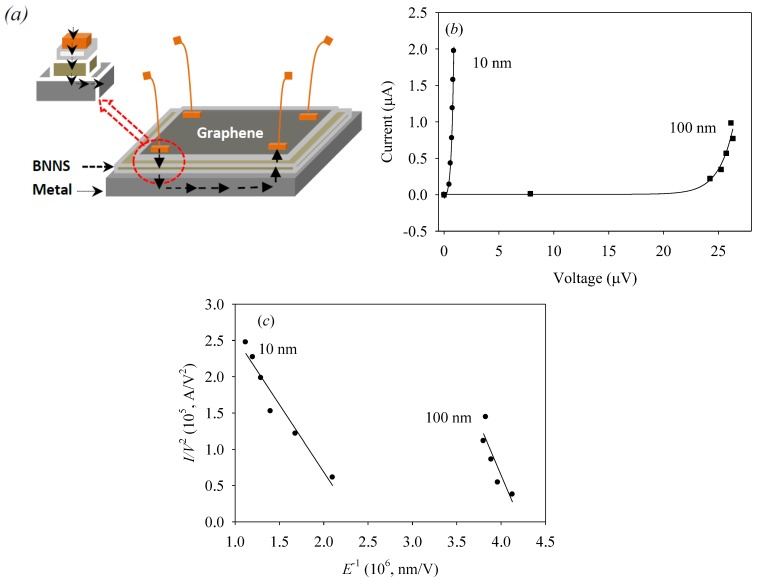
(**a**) Schematic illustration of G/BNNS/metal (G/BNNS/M) heterostructure tunneling device structure. (**b**) Schottky barrier characteristics measured for two G/BNNS/M devices having different BNNS thickness, curve 1 (10 nm thick BN film) and curve 2 (100 nm thick BN film). (**c**) Flower-Nordheim plot for different thickness of BNNS, circle curve shows plot for ~10 nm thickness BNNS and square plot belongs to ~100 nm thick BNNS.

**Figure 4 nanomaterials-09-00925-f004:**
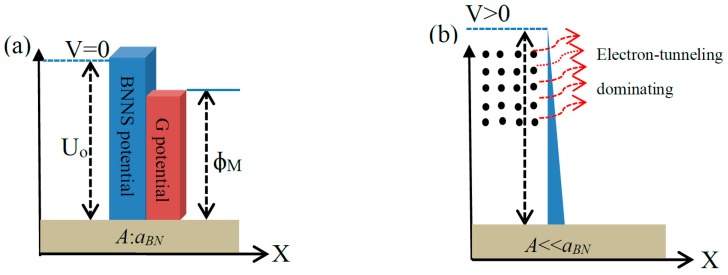
(**a**) Schematic representation of potential barriers having arbitrary high for G/BNNS heterostructures. (**b**) Electron tunneling through the modified G/BNNS potential barrier width at applied V>0.

**Figure 5 nanomaterials-09-00925-f005:**
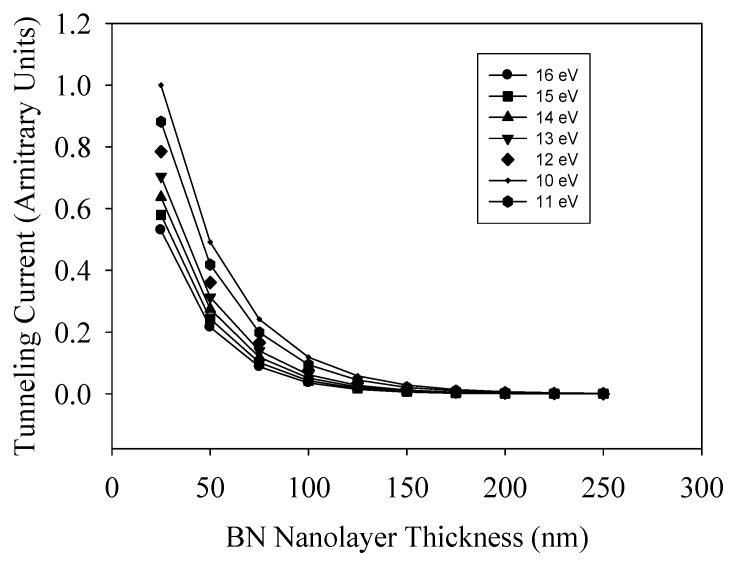
Dependence of relative tunneling current on the BNNS layer thickness with different effective potential amplitudes.

**Figure 6 nanomaterials-09-00925-f006:**
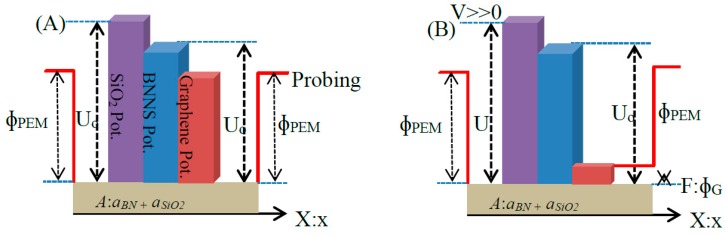
Schematic representation of potential barriers for (**a**) G/BNNS/SiO_2_ and (**b**) PEM/G/BNNS/SiO_2_ systems.

**Table 1 nanomaterials-09-00925-t001:** Graphene layer sheet resistance measurements for different combinations of heterostructures.

Heterostructure	R_s_ (Ω/sq.)
G/SiO_2_	(1.37 ± 0.15) × 10^3^
G/BNNS/SiO_2_	(6.03 ± 0.94) × 10^6^
G/BNNS/Mo	(3.67 ± 5.60) × 10^−4^

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
