# Peer review of "Synthesis, Characterization and Fabrication of Graphene/Boron Nitride Nanosheets Heterostructure Tunneling Devices"

_nanomaterials, 2019, doi:10.3390/nano9070925_

Reviewer 1 Report

This paper deals with the fabrication and electrical characterization of graphene (GG) heterostructrures with boron nitride nanosheets (BNNS) on a metal substrate, with the aim of realizing tunneling devices.

BNNS were deposited by pulsed laser deposition on a metal (M) substrate or onto SiO2.

Gr was grown by CVD on copper and transferred onto BNNS/SiO2, BNNS/Cu and SiO2.

Sheet resistances in the vdP configuration were evaluated for the G/BNNS/SiO2, G/BNNS/M and G/SiO2 samples, showing a very low sheet resistance for the G/BNNS/M heterostructure. This was explained by current tunneling from G to Mo through the BNNS layer.

The following major point should be addressed before the paper can be considered for publication:

1.     What is the reason of the 3 orders of magnitude higher sheet resistance measured on the G/BNNS/SiO2 sample with respect to the G/SiO2 sample? I would have expected to find similar values of sheet resistance for these two samples, where G is on top of an insulator.

2.     The thickness of BNNS in the two G/BNNS/M samples considered in Fig.3 must be provided.

Furthermore, the thickness uniformity of the BNNS should be discussed, as small local thickness variations can result in large current changes.

3.     The voltage axis scale in Fig.3b is not clear. Is the voltage in the 1e-5 V range? This is different than what reported in the main text.

4.     The current tunneling model proposed by the authors is indeed not new. Basically, they derive the expression of Fowler-Nordheim (FN) tunneling through a triangular barrier.

I would have expected that the authors report a fit of the experimental data in Fig.3b with a Fowler-Nordheim plot, to extract the barrier height at Gr/BNNS interface.

Furthermore, current-voltage characterization at different temperatures should be performed to support the FN tunneling mechanism (nearly independent on the temperature)

5.     What is the meaning of tunneling current in arbitrary units in Fig.5?

6.     A more general overview of recent development on vertical 2D heterostructures devices should be provided in the introduction (see, as a reference, Crystals 8 (2), 70 (2018)).

Author Response

Dear Editor and Reviewer

We would like to thank the Editor and reviewers for their careful review of our manuscript and constructive suggestions and comments to improve the quality of the paper. We have carefully considered reviewers suggestions and have done our best to address all the points in the revised manuscript.

Below is a point-by-point response to the reviewers’ comments and suggestions.

Review Report 1:

Q1.     What is the reason of the 3 orders of magnitude higher sheet resistance measured on the G/BNNS/SiO2 sample with respect to the G/SiO2 sample? I would have expected to find similar values of sheet resistance for these two samples, where G is on top of an insulator.

A1. The measured resistance difference was explained by different scales of interaction between G and SiO2 surfaces, and between G and BNNS/SiO2, respectively. It has been reported in the literature that the BNNS are free from dangling bonds, have smooth surface with similar lattice constant to that of graphene thus inducing small perturbation in graphene, and hence reduce its sheet resistance. The more obvious effect in low sheet resistance in graphene is due to the vertical charge tunneling through the BNNS. However, in the case of G/BNNS/SiO2, the insulating nature of BNNS/SiO2 substrate such as 2D/3D gives large thickness which reduce the charge tunneling probability, hence sheet resistance reduced enormously.

Moreover, in the second case of G/BNNS/SiO2, the sample and substrate interaction should signifyingly be larger than in the first case, because BN crystal structures are close in comparison with G and SiO2 crystal structures, where structures strongly differ. Such interaction perturbs G zone structure, and induces increase of the respective band gap (E) between valence and G crystal first conductive band. Taking into account that, sheet resistance can approximately be considered proportional to e-(E/kT) factor. Thus, to observe difference of three orders in resistance for the respective samples, difference between energy gaps for both samples of interest should be around ~7kT which corresponds to1400 cm-1 at room temperature. To further substantiate this estimate one shall carry out detailed ab initio calculation for G zone structures in both discussed cases. However, such analysis is not a goal of the current work, and will be conducted independently.  

Q2.     The thickness of BNNS in the two G/BNNS/M samples considered in Fig. 3 must be provided.

Furthermore, the thickness uniformity of the BNNS should be discussed, as small local thickness variations can result in large current changes.

A2. We agree with the reviewer’s comment that the small local thickness variation can result in large current change. However, this is not the case in the present study. In our earlier work [19,20] we optimized the experimental conditions for the synthesis of BNNS films with different thicknesses. Here, similar deposition method and growth conditions were applied for the synthesis of BNNS. Using optimized experimental parameters, we can successfully grow BNNS thin films with top layer composed of uniform sheet with 20×20 µm grain boundary which was sufficient to construct and study device characteristics. Please note that the results shown in this paper are from the part of the samples we transfer from the deposited films to TEM grid.

In revised manuscript, we represent averaged film thickness estimated using XRD method and algorithm described earlier by M. Yasaka, Rigaku J. 26, 1–9 (2010). We have added this reference in our revised manuscript as [23]. The estimated thickness values are around 9.79±1.36 nm and 101.23±3.61 nm, respectively, for samples selected for I/V characteristic measurement shown in Fig. 3b.

Q3.     The voltage axis scale in Fig. 3b is not clear. Is the voltage in the 1e-5 V range? This is different than what reported in the main text.

A3.  We upgraded the text in our revised manuscript.

Q4.     The current tunneling model proposed by the authors is indeed not new. Basically, they derive the expression of Fowler-Nordheim (FN) tunneling through a triangular barrier.

I would have expected that the authors report a fit of the experimental data in Fig.3b with a Fowler-Nordheim plot, to extract the barrier height at Gr/BNNS interface.

Furthermore, current-voltage characterization at different temperatures should be performed to support the FN tunneling mechanism (nearly independent on the temperature)

A4. We agree with Referee comment. We used developed earlier tools for analysis of our experimental data and extended it for triangular barrier. The expression of Fowler-Nordheim (FN) tunneling through a triangular barrier is the key point to understand barrier tunneling through G/BNNS. Also, this is an interested suggestion to perform current-voltage characteristics at different temperature to support FN tunneling mechanism. Unfortunately, at present we are unable to conduct such measurements from our device due to lack of experimental facilites available. However, we will consider this suggestion for extended set of samples and will report in the next study.

We included averaged value of the fitting parameter U0 in revised manuscript (see p.8). We tested the model sensitivity to the U0 variation around ionization threshold of BN material to test the tunneling current dependence on the barrier high and barrier thickness.

Q5.     What is the meaning of tunneling current in arbitrary units in Fig.5?

A5.  It is the ratio Itunn/Itunn,max.

Q6.     A more general overview of recent development on vertical 2D heterostructures devices should be provided in the introduction (see, as a reference, Crystals 8 (2), 70 (2018)).

A6. We upgraded the introduction part and cited the reference in our paper.

At the end, we once again thank to editor and all the reviewers for their valuable feedback on our paper.

Best regards

Muhammad Sajjad, PhD

Reviewer 2 Report

please see attached.

Author Response

Review Report 2:

Dear Editor and Reviewer

We would like to thank the Editor and reviewers for their careful review of our manuscript and constructive suggestions and comments to improve the quality of the paper. We have carefully considered reviewers suggestions and have done our best to address all the points in the revised manuscript.

Below is a point-by-point response to the reviewers’ comments and suggestions.

Q1. The introduction is confusing for its unsupported use of nomenclature (What is a “single material composite”? “…stabilizing the AA’ stacking mode”?) and seeming contradictions (describing BNNS as insulating and conductive). The introduction needs rewriting so it clearly states the needed technical background and intent for the current paper.

 A1. We modified the introduction part and highlighted device characteristics.

“Single material composite” means a single 2D/2D materials heterostructure.

“stabilizing the AA’ stacking mode”, means that it is expected that the polarity of the B-N bond results in inter-layer electrostatic interactions with the C atoms residing on the underlying B or N atoms, thus stabilizing the AA' stacking mode [4].

“Contradictions (describing BNNS as insulating and conductive)”, in general BNNS are wide bandgap semiconductor material with band gap up to 6eV indicating strong insulating properties of this material. However, out-of-plane nanolayer BNNS exhibits high charge tunneling phenomenon that describes its importance as tunneling device. The in-plane insulating properties and out of plane conductivity of BNNS was reported in the literature [Liam Britnell et. al. “Electron Tunneling through Ultrathin Boron Nitride Crystalline Barriers”, Nano Lett.20121231707-1710].

We have revised manuscript to make it less ambiguous and easy to understand for a reader. We have also added more device characteristics in the revised manuscript.

Q2. The “device” part of the paper is hampered by a lack of data. Two I-V characteristics are shown with poorly defined axes. I am assuming that the unit used in line 181 is mV? The authors do not describe their measurements for other devices. For example, does each metal contact on the graphene for the Van der Pauw measurements give identical I-V characteristics through the layers? How many devices of different BNNS layer thicknesses were measured? The paper give the impression that only two devices (one thin BNNS, one thick BNNS) were actually measured. This is not sufficient to report. 

A2. We agree with the reviewer comments. Infect, in our study we tested four devices independently. Three devices were composed of G/BNNS/M with different thickness of BNNS, 10 nm, 100 nm and few microns BNNS sandwiched between graphene and metal substrate. Two devices with BNNS 10 nm and 100 nm have shown excellent IV behavior and data is presented in current paper. The device prepared with few micron BNNS thickness (~ 6µm) did not show IV characteristics due to large barrier thickness, and lack of tunneling mechanism. Similar situation was observed in the case of G/BNNS/SiO2 and we also did not present data in our study. Reason is G/BNNS/SiO2 did not show IV characteristics due to the high resistance of the device as two nonconducting surface were in contact underneath of graphene layer.

We updated this information in revised manuscript. The measurements conducted for G/BNNS/M shown IV characteristics Curve 1 with 10 nm and curve 2 at 100 nm BNNS thickness as in Fig. 3(b). During the measurements, we slightly displaced metal probe contacts and did not observe any significant change in the IV characteristics of the device which shows uniformity of the BNNS film and the systematic behavior of the device. 

Q3. The modeling part is somewhat convoluted which seems a direct result of the lack of data to model. A set of measurements on devices with different known BNNS thicknesses that shows an exponential decrease in current with thickness would easily make the argument for tunneling transport. (This is the “classic” signature of the tunneling mechanism.)

A3. Thank you for this comment. We missed this statement in our consideration, and in revised text, we included sentence: Decrease of current with increase of BNNS thickness is direct prove of the electron tunneling transport mechanism. Here, we would like to mention that we also tested one device with BNNS thickness up to few microns. We did not see any IV dependences indicating that increase thickness of BNNS decrease or completely stop vertical tunneling current. 

At the end, we once again thank to editor and all the reviewers for their valuable feedback on our paper.

Best regards

Muhammad Sajjad, PhD

Round  2

Reviewer 1 Report

Unfortunately, authors did not addressed in a satisfactorily way the main concerns raised in my first report. Hence, I do not feel this paper can be accepted for publication in Nanomaterials.

In the following, my detailed comments:

1.     The authors’ explanation for the large difference in the sheet resistance measured on the G/BNNS/SiO2 sample with respect to the G/SiO2 sample seems to me confusing and not convincing. They tried to explain this difference by the different scales of interaction between G and SiO2 surfaces, and between G and BNNS/SiO2. Firstly they mentioned that the BNNS are free from dangling bonds, have smooth surface with similar lattice constant to that of G thus inducing small perturbation in G, and hence reduce its sheet resistance. Afterwards, they wrote that, because the close crystal structures of G and BN, the G/BNNS interaction should be larger than for G/SiO2, resulting in the presence of a bandgap E for G on BN, that could explain the larger sheet resistance.

Indeed, my opinion is the authors should provide further evidence (e.g. Raman spectra) that the structural quality of G transferred onto BNNS/SiO2 is the same as that of G onto SiO2. The higher sheet resistance for the G/BNNS/SiO2 sample can be simply due to a bad quality of G, e.g. due to damage introduced during transfer.

2.     The authors did not report any fit of the I-V curves in Fig.3b with a Fowler-Nordheim plot.

Author Response

Dear Editor

We would like to thank the reviewers for their careful review of our manuscript (Synthesis, Characterization and Fabrication of Graphene/Boron Nitride Nanosheets Heterostructure Tunneling Devices) and constructive suggestions and comments to improve the quality of the manuscript. Enclosed you will find the revised article (2nd revision), addressing the reviewers’ comments and recommendations. This version comprises all the required amendments made based on the suggestions and remarks made by the reviewers. We have also performed an overall revision of grammatical constructions and typos in order to improve the manuscript’s readability. The authors’ detailed response to reviewers’ comments and changes made to the article are detailed below:

Reviewer 1, Comment 1: The authors should provide further evidence (e.g. Raman spectra) that the structural quality of G transferred onto BNNS/SiO2 is the same as that of G onto SiO2. The higher sheet resistance for the G/BNNS/SiO2 sample can be simply due to a bad quality of G, e.g. due to damage introduced during transfer.

Answer: We added the following statement on P4 in the revised manuscript to explain the quality control protocol implemented: “The quality of the graphene sheets was verified at various stages along the process using Raman spectroscopy: as-grown, on PMMA, and after transfer to SiO2 and BNNS. The devices that ended up with poor quality graphene sheet due damage/contamination during the transfer process were discarded.”

Reviewer 1, Comment 2:  The authors did not report any fit of the I-V curves in Fig.3b with a Fowler-Nordheim plot.

Answer: We have now included the Fowler-Nordheim plot in Fig. 3c and added the following sentence to the manuscript on P7 “The observed decrease in current with increase of BNNS film thickness (Fig. 3(c)) was analyzed using the Fowler-Nordheim formalism finding that it is consistent with the electron tunneling.

Figure: Flower-Nordheim plot for different thicknesses of BNNS; circle curve shows plot for ~10 nm thickness BNNS and square plot belongs to ~100 nm thick BNNS.

Reviewer 2 Report

See attached.

Author Response

Dear Editor

We would like to thank the reviewers for their careful review of our manuscript (Synthesis, Characterization and Fabrication of Graphene/Boron Nitride Nanosheets Heterostructure Tunneling Devices) and constructive suggestions and comments to improve the quality of the manuscript. Enclosed you will find the revised article (2nd revision), addressing the reviewers’ comments and recommendations. This version comprises all the required amendments made based on the suggestions and remarks made by the reviewers. We have also performed an overall revision of grammatical constructions and typos in order to improve the manuscript’s readability. The authors’ detailed response to reviewers’ comments and changes made to the article are detailed below:

Comment 1: Out-of-plane nanolayer BNNS exhibits high charge tunneling phenomenon that describes its importance as tunneling device. The in-plane insulating properties and out of plane conductivity of BNNS was reported in the literature [Liam Britnell et. al. “Electron Tunneling through Ultrathin Boron Nitride Crystalline Barriers”, Nano Lett.20121231707-1710]. This information needs to be added to the last paragraph in the introduction and the reference given.

Answer: This reference by Liam Britnell et al is now included as Ref. 13 and the following sentence was added to the manuscript on P2 “Nanothin BN, such as our BNNS, shows in-plane insulating properties and out-of-plane charge tunneling phenomenon [13]. The out-of-plane electron tunneling through atomically thin layers of BNNS makes this material advantageous for 2D/2D heterostructure devices where BNNS can be used as gate-controlled p-layer [13].”

[13]      Britnell L; Gorbachev RV; Jalil R; Belle BD; Schedin F; Katsnelson MI; Eaves L; Morozov SV; Mayorov AS; Peres NMR; Castro Neto AH; Leist J; Geim AK; Ponomarenko LA;  Novoselov KS; Electron Tunneling through Ultrathin Boron Nitride Crystalline Barriers; Nano Lett. 2012, 12, 1707.

In general, when describing BNNS bulk properties and monolayer or few layer BNNS properties may be confusing when one call bulk BNNS as insulating material and monolayer or few layer BNNS as conductive due to its out-of-plane charge tunneling mechanism. Bulk BNNS is an insulator (dielectric material) with bandgap of about 6 eV. However, if thin BNNS layer is deposited on the metal surface, significant tunneling charge transport through BNNS between electrodes is observable which may to ambiguity whether to call it an insulator or a conducting material. We restricted our self not to use word “conductivity” directly for BNNS but at the same time we cannot ignore the out-of-plane charge tunneling mechanism in thin BNNS as this is the key part of this study. We have revised manuscript accordingly by eliminating the word “conductivity” in Introduction section and clarified the relevant statement.

Q2a. My questions about “…does each metal contact on the graphene for the Van der Pauw measurements give identical I-V characteristics through the layers?” is not answered. 

A2a. Sorry if we miss this point to highlight in our previous response. Yes, all metal contact electrodes pairs used for I/V characterization have shown identical results.

Q2b. The IV data in Figure 3b supposedly comes from a measurements between electrode 1 and electrode 4. Nominally, one would expect a similar IV between electrode 2 and electrode 3. 

A2b. We agree with the reviewer’s comment. Electrodes 2 and 3 were also showing similar IV measurements.

Q2c. In the authors response is the sentence “During the measurements, we slightly displaced metal probe contacts and did not observe any significant change in the IV characteristics of the device which shows uniformity of the BNNS film and the systematic behavior of the device.” Does this indicate that the contacts, and thus the gold colored electrodes in the schematic, were made by movable metal probes? What is meant by “slight displacement”?

A2c. We agree that using the “slight displacement” was unfortunate and confusing. The contacts on the surface of the tested samples were permanently fixed. However, what we meant by “slight displacement” was that the external probes could be connected at any point on top of electrodes. We performed multiple measurements using the same set of metal contacts by touching the external probes at different point on top of a single electrode to eliminate any possible variation between individual IV measurements. 

We have fixed the language in our revised manuscript on P7 as follows: “The observed decrease in current with increase of BNNS film thickness (Fig. 3(b)) was analyzed using the Fowler-Nordheim formalism finding that it is consistent with the electron tunneling transport mechanism described above.”

Q2d. The information the authors provide in their answer (A2) to the first review (Q2) needs to be discussed in the actual text of the paper, not just in the response to the reviewer to inform the reader appropriately. Given the minimal amount of “device” data the authors present, it is necessary for them to give a more complete description of the measurements that were made. 

A2d. We have modified the text accordingly by adding requested info on measurement specifics on P4 in the experimental section of the manuscript.

Q3. Concerning the tunneling mechanism, the authors have added the sentence “Decrease of current with increase of BNNS thickness is direct prove of the electron tunneling transport mechanism.” 

I am sure the authors’ intended to use the word “proof” as opposed to “prove”, but this is too strong a word. I agree that the two thicknesses give “direct evidence” for tunneling. I suggest the authors use this wording.

A3. Thanks for pointing out this mistake. We have corrected it in our revise manuscript.

Round  3

Reviewer 1 Report

The authors addressed most of the referees' comments.

Paper can be accepted for publication

Reviewer 2 Report

I accept the author's revisions.